# Protocol for the crowdsourced image-based morbidity hotspot surveillance for neglected tropical diseases (CIMS-NTDs)

**Uchechukwu Madukaku Chukwuocha**[ID]*, **Christopher Sule Oyamienlen, Ayoola Oluwaseun Bosede**[ID]**, Ikechukwu Nosike Dozie**

Innovations and Technologies for Disease Control Research Group, Department of Public Health, Federal University of Technology, Owerri, Nigeria

* uchukwuocha@gmail.com

## Abstract

**Data Availability Statement:** No datasets were generated, as this manuscript only describes the protocol for a new method.

## Introduction

Efficient NTDs elimination strategies require effective surveillance and targeted interventions. Traditional methods are costly and time-consuming, often failing to cover entire populations in case of movement restrictions. To address these challenges, a morbidity image-based surveillance system is being developed. This innovative approach which leverages the smartphone technology aims at simultaneous surveillance of multiple NTDs, enhancing cost-efficiency, reliability, and community involvement, particularly in areas with movement constraints. Moreover, it holds promise for post-elimination surveillance.

## Methodology

The pilot of this method will be conducted across three states in southern Nigeria. It will target people affected by Neglected Tropical Diseases and members of their communities. The new surveillance method will be introduced to target communities in the selected states through community stakeholder's advocacy meetings and awareness campaigns. The pilot which is set to span eighteen months, entails sensitizing NTDs-affected individuals and community members using signposts, posters, and handbills, to capture photos of NTDs manifestations upon notice using smartphones. These images, along with pertinent demographic information, will be transmitted to a dedicated server through WhatsApp or Telegram accounts. The received images will be reviewed and organized at backend and then forwarded to a panel of experts for identification and annotation to specific NTDs. Data generated, along with geocoordinate information, will be used to create NTDs morbidity hotspot maps using ArcGIS. Accompanying metadata will be used to generate geographic and demographic distributions of various NTDs identified. To protect privacy, people will be encouraged to send manifestation photos of the affected body part only without any identifiable features.

**Funding:** UMC received the grant. The grant number is INV-048721. The funder is Bill and Melinda Gates Foundation https://www.gatesfoundation.org/ The funder did not play any role in the design, analysis, decision to publish or preparation of the manuscript.

**Competing interests:** The authors have declared that no competing interests exist.

## Evaluation protocol

NTDs prevalence data obtained using conventional surveillance methods from both the pilot and selected control states during the pilot period will be compared with data from the CIMS-NTDs method to determine its effectiveness.

## Expected results and conclusion

It is expected that an effective, privacy-conscious, population inclusive new method for NTDs surveillance, with the potential to yield real-time data for the identification of morbidity hotspots and distribution patterns of NTDs will be established. The results will provide insights into the effectiveness of the new surveillance method in comparison to traditional approaches, potentially advancing NTDs elimination strategies.

## Introduction

Neglected Tropical Diseases (NTDs) constitute a group of debilitating and often overlooked diseases that primarily affect the world's most impoverished populations [1,2,3]. These diseases encompass a diverse range of conditions, including Lymphatic filariasis, Onchocerciasis, Schistosomiasis, Soil-transmitted helminthiasis, and Trachoma, to name a few. They share a common characteristic in their tendency to thrive in marginalized communities with limited access to basic healthcare, clean water, and sanitation facilities. The global burden of NTDs is staggering, with an estimated 1.7 billion people affected, many of whom live in low- and middle-income countries [1]. These diseases not only cause immense suffering but also perpetuate the cycle of poverty, hindering economic development and preventing affected individuals from reaching their full potentials. The consequences extend beyond health, impacting education, workforce productivity, and overall societal well-being [4].

Some NTDs have been designated as targets for elimination by the World Health Organization (WHO) in the NTDs Road Map 2021–2030 [5]. Some are targeted for eradication (Dracunculiasis and Yaws), some are targeted for control while others are targeted for elimination as a public health problem. In 2022, the Kigali Declaration on NTDs was issued to galvanize political support and secure commitments aimed at achieving Sustainable Development Goal 3 (SDG3: Good Health and Well-being) targets related to NTDs and aligning with the objectives outlined in the WHO's Neglected Tropical Disease Roadmap 2021–2030 [5].

The WHO's NTDs Road Map 2021–2030 outlines a strategic framework for the control, elimination, and eradication of these diseases, with the ultimate goal of achieving a world free of NTDs-related suffering by the end of the decade [3]. Effective surveillance and precise intervention strategies are indispensable components in the pursuit of NTDs elimination [6,7]. While significant progress has been made in recent years, considerable challenges persist. Traditional surveillance methods, often reliant on physical visit of healthcare workers to communities, are resource-intensive and geographically limited. This limitation has been particularly evident during such crises as the COVID-19 pandemic, civil unrest, and natural disasters, when access to affected regions becomes severely restricted [8]. Also, Nigeria lacks an active surveillance system for NTDs, relying instead on passive reporting. This passive approach involves generating NTDs prevalence data when cases happen to present themselves at healthcare facilities for complaints or treatment. Following diagnosis confirmation, the cases are reported to Local Government Area Disease Surveillance and Notification Officers (DSNO),

who subsequently forward the reports to State-level DSNOs through the Integrated Disease Surveillance and Response System (IDSR) [7]. However, this passive surveillance method falls short in supporting the mapping of NTDs prevalence required for targeted interventions, aligning with the global goal of eliminating these diseases by 2030. It also does not support timely report of cases as well as the active involvement of community members in the surveillance process. Furthermore, many NTD cases in endemic areas go unreported at healthcare facilities due to the stigma associated with these conditions and the perceived shame they may bring upon those affected.

Addressing these challenges requires innovative solutions that can augment existing surveillance efforts, reach underserved populations, and provide timely data for decision-making. Notably, recent experiences have demonstrated that crowdsourced data-based surveillance methods offer a cost-effective, real-time, scalable, and efficient means to enhance malaria surveillance [6]. It is within this context that this proposed project seeks to introduce a transformative approach to NTDs surveillance—one that leverages the power of crowdsourcing, smartphone technology, and artificial intelligence to create a dynamic and responsive system capable of delivering real-time insights into the distribution and prevalence of NTDs. By this, it is aimed that the global effort to combat NTDs will be bolstered, offering a cost-effective and scalable alternative that can be integrated into ongoing programs and bring the world closer to the realization of a world free from the burden of NTDs. Building on these insights, the proposed project aims to make a substantial contribution to achieving the WHO's NTDs elimination targets. We therefore proposed the development and validation of a Crowdsourced Image-Based Morbidity Hotspot Surveillance Method for Neglected Tropical Diseases (CIMS-NTDs). This will be realized as follows;

First, an electronically transmitted crowdsourced photograph method to enhance NTDs surveillance will be created and validated. To achieve this, we will educate individuals with NTDs manifestations and their communities on using their smartphones to capture images of these manifestations. These images, along with essential demographic information, will be transmitted via WhatsApp and Telegram to a designated mobile telephone number. Subsequently, the collected data will be organized in an electronic database, where experts will identify and categorize the images by specific NTDs. Geolocation data will be leveraged to create morbidity hotspot/ distribution maps for these NTDs, which will aid policymakers in effective planning and targeted deployment of interventions. Importantly, this method also has the potential to support post-elimination surveillance by enabling swift reporting of potential cases by the empowered community members.

Secondly, we aim to develop a feedback mechanism and personalized care for identified NTDs subjects. Utilizing the mobile telephone numbers through which the images were submitted, personalized feedback would be provided on preliminary diagnosis, counselling on immediate morbidity management, and disability prevention to the individual subjects. This also offers non-clinical support on hygiene practices (care and management of elephantiasis, for instance), transmission avoidance behaviors, and guidance on accessing interventions. Subjects whose photographs were sent on their behalf by our CCPs will also receive their feedback through the CCPs who will be trained on basic NTDs management counselling and feedback provision.

Finally, we aim to develop and train an Artificial Intelligence (AI)/Machine Learning (ML) algorithm, along with a mobile application. This tool, utilizing categorized images and NTDs knowledge, will aid healthcare personnel in differential diagnosis and offer real-time clinical management or referral advice at health facilities. This AI/ML model aims to replace human annotators for swift and more accurate identification and matching of manifestation photographs to specific NTDs. It will eliminate time lag and potential errors associated with human

annotation as well as enhance NTDs differential diagnosis, aiding front-line health workers in their duties.

## Materials and methods

### Study setting

The pilot of the Crowdsourced Image-Based Morbidity Hotspot Surveillance Method for Neglected Tropical Diseases (CIMS-NTDs) will be carried out in Nigeria. Nigeria has a tropical climate with variable rainy and dry seasons, depending on location. It is hot and wet most of the year in the southeast but dry in the southwest and farther inland. Nigeria is one of the most densely populated nations in Africa, the most populous country in the African continent, and the seventh-most populous nation in the world [9]. Hence, Nigeria is a likely target market for mobile phone, internet, and telecommunication businesses, and as such has influenced many Nigerians to becoming smartphones owners. It is estimated that about 90% of the country's population are mobile phone owners [10]. Nigeria has a significant amount of cultural diversity, due to its more than 250 distinct ethnic groups, more than 500 languages, and the variety of traditions practiced by each of them. Nigeria's economic situation is such that more than 40% of the population lives in poverty, with poor social and healthcare infrastructure [11]. This situation together with environmental and climatic factors encourage the spread of a variety of diseases, including neglected tropical diseases. Nigeria bears the second highest global burden of NTDs [12]. At least one NTD is endemic in all the 36 states plus the Federal Capital Territory in Nigeria.

### Study design

**Sample size, inclusion and exclusion criteria.**   The study will be conducted in six (6) states in Nigeria. These pilot states will be purposively selected based on available data on the endemicity of the target NTDs from Nigeria's Federal Ministry of Health and Social Welfare. Three (3) of these states will be randomly selected as the intervention states where the proposed CIMS-NTDs will be implemented, while the other three states will serve as controls where the CIMS-NTDs will not be implemented. However, NTDs prevalence data recorded during the study period in these control locations through the traditional NTDs surveillance method will be assessed and documented. An additional criterion for selection will be, endemic states where the Partner Non-Governmental Organizations (NGOs) are already operating. NTDs surveillance data will be collected from State NTDs programme offices of both the intervention and control states. These data will be used to compare outcomes between the CIMS-NTDs method and the conventional method within the intervention states as well as the control states. The study will be a population-based study, and every available member of the project implementation sites shall be recruited into the study.

**Description of processes.**   Fig 1 is the flowchart of the entire project process, while Fig 2 is a simple pictorial depiction of the CIMS-NTDs concept. The project process by timeline is presented as follows;

*Inception meeting of project partners (Month 1).* This project will commence with a hybrid meeting (virtual and in-person) of all project partners. The project partners include the Federal Ministry of Health and Social Welfare of Nigeria Programme Offices (NPO) for NTDs, and NGOs- whose role will be majorly community mobilization, sensitization and follow up. This meeting will involve further familiarization of project partners, assessment of the project design, procedures and protocols, confirmation of assigned roles as well as setting up the work flow/system for the project.

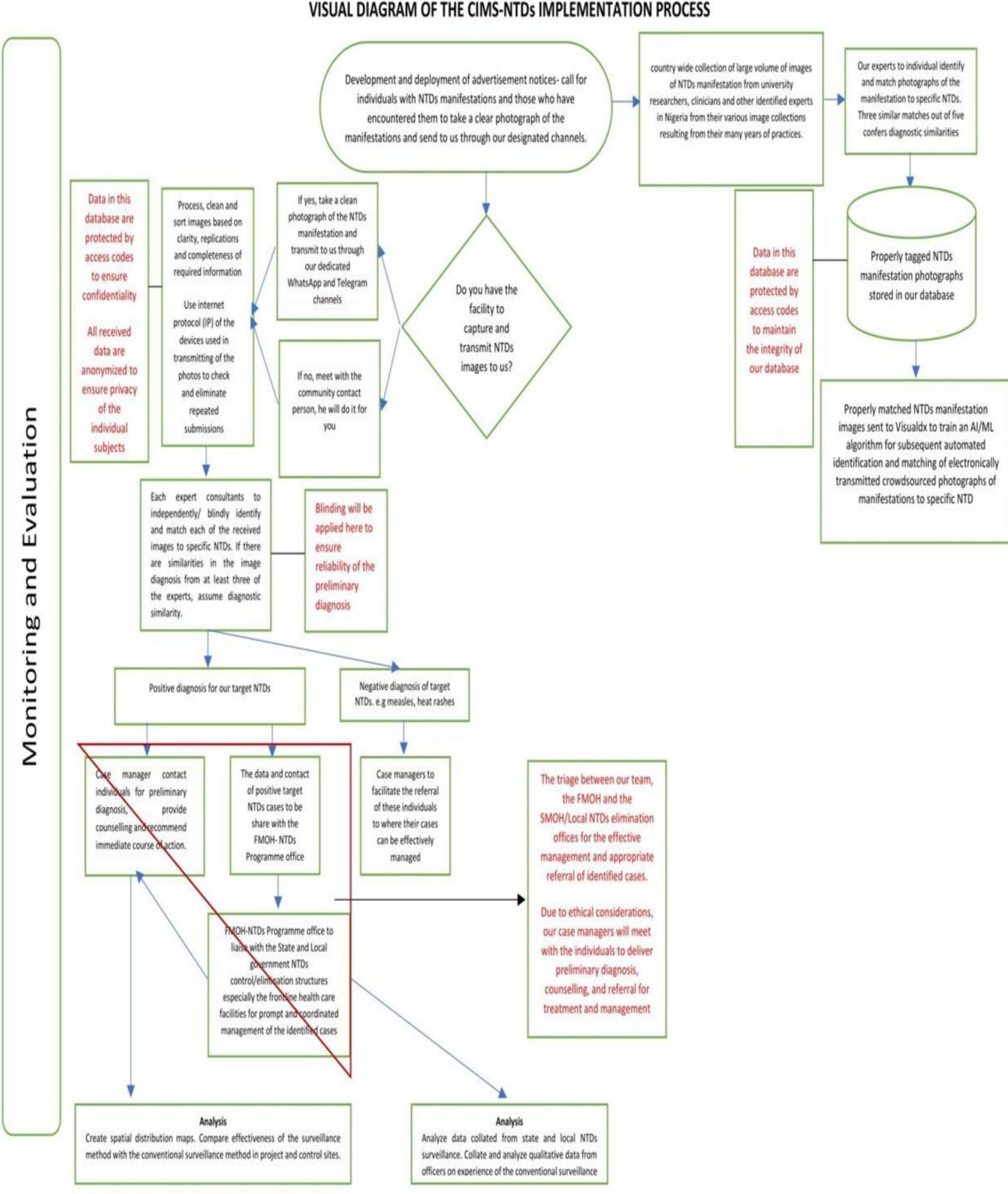

**Fig 1. Flow diagram of the CIMS-NTDs process.**

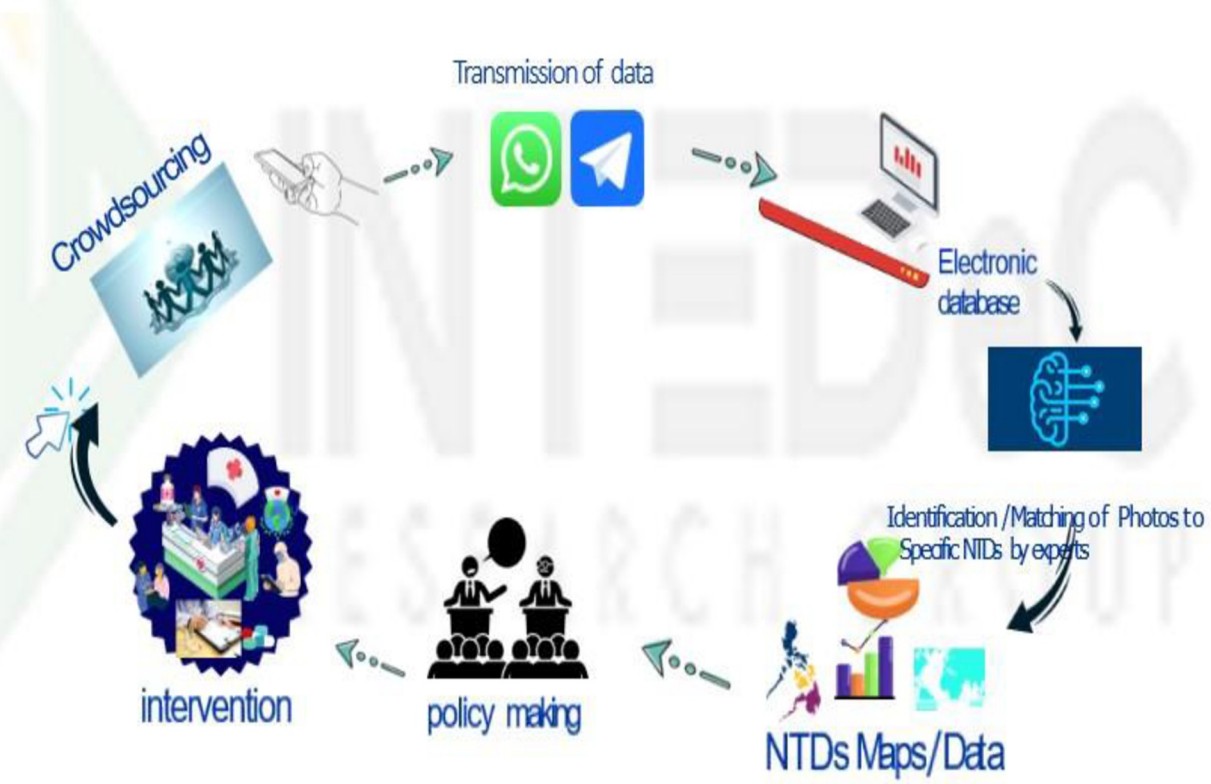

**Fig 2. The simple pictorial depiction of the CIMS-NTDs process.**

*Advertisement design and project sites selection (Month 1).* At the beginning of the project, the process of designing the project advertisement and selection of the project sites will be initiated. This crucial phase involves the following key steps:

**1. Photo and literature search:** We will conduct an extensive search for photographs depicting the manifestations of the targeted Neglected Tropical Diseases (NTDs), including Onchocerciasis, Lymphatic filariasis, Trachoma, Buruli Ulcer, Guinea worm infection, Leprosy, and Ascariasis. These photos will serve as visual references for our awareness campaign.

**2. Photo validation:** Selected photos will undergo validation by National Programme Office for NTDs officials, following the World Health Organization (WHO) standard operating procedures (SOP). This validation process ensures the accuracy and relevance of the selected images.

**3. Advertisement content:** The validated photos will be used to craft our study advertisement/notice. The content of the advertisement will include:

- Distinctive photos illustrating the manifestations of the targeted NTDs, accompanied by descriptions in both English and local languages relevant to the study locations.

- A call to action for individuals who either have NTDs manifestations or have encountered them to capture clear photographs of these manifestations using their smartphones. Consent from the subjects will be obtained, and the images will exclude the subjects' faces and other indefinable features.

- Instructions to transmit the photographs, along with demographic information about the subject (including location, age, gender, occupation, marital status, and educational level),

through designated social media channels such as WhatsApp and Telegram. Alternatively, community members lacking the necessary technology can utilize a designated community contact person (CCP), whose name and telephone number will be provided on the advertisement.

- A notice indicating that the NPO will provide available intervention measures to those who comply with the request.

**4. Multilingual narratives:** Advertisements will feature narratives in both English and the local languages spoken in the pilot locations to ensure broad accessibility and understanding.

**Advertisement materials:** To disseminate this critical information effectively, we will produce the following advertisement materials:

- Signposts designated as *C2SSS* (Community Crowdsourced Surveillance Sensitization Signposts): These 8ft by 4ft signposts will be strategically placed in each study community.

- Posters designated as *C2SSP* (Community Crowdsourced Surveillance Sensitization Posters): Approximately 300 posters will be affixed in and around each pilot community to maximize visibility.

- Flyers designated as *C2SSF* (Community Crowdsourced Surveillance Sensitization Flyers): Around 1000 flyers will be distributed throughout the pilot communities, with an additional

300 flyers shared every three months to maintain continued awareness within the communities.

This phase sets the stage for community engagement, data collection, and the implementation of our crowd-sourced image-based morbidity surveillance method.

## 5. Selection of pilot study sites

To ensure a robust and comprehensive study, we will follow a systematic process to select our pilot study sites;

### a. Choice of endemic states

We will begin by identifying states with high prevalence of Neglected Tropical Diseases (NTDs) based on data provided by the Federal Ministry of Health and Social Welfare (FMOH) NTDs Programme Office.

### b. Study site categories

**Project implementation sites:** Among the identified states, three will be designated as project implementation sites, where we will actively implement and evaluate our crowdsourced image-based morbidity hotspot surveillance method.

**Control Sites:** The remaining three states will serve as control sites. In these states, we will obtain the usual NTDs surveillance data that will later be used as a basis for comparison with the data generated by the piloted surveillance method.

### c. Selection of local government areas (LGAs)

In the selected project implementation states, we will further narrow down our focus by selecting five Local Government Areas (LGAs) from each state. These LGAs will be selected based on recommendations and NTDs prevalence data from the respective state's NTDs programme office.

### d. Community/ward selection

From each of the five selected LGAs, we will equally select three communities/wards to serve as our project pilot study sites.

This meticulous selection process allows us to gather data from both project implementation sites and control sites, ensuring a comprehensive assessment of the CIMS-NTDs' effectiveness. By comparing data from the two groups, we aim to provide valuable insights into the effectiveness and potential benefits of our innovative approach.

*Advocacy visits, engagement of local stakeholders and inception workshop (Month 2)*. Stakeholders in the selected states, including officials from State and Local Government health authorities, religious and community leaders, as well as local and international NGOs supporting NTDs programmes in the intervention states, will be engaged to solicit their support in sensitizing and mobilizing community members, and in any other way that could help the project succeed.

Community Contact Persons (CCPs) will be selected in each pilot community by the community members, preferably from among suitable Community Directed Distributors (CDDs) of ivermectin or any suitable community member who can operate a smartphone. These individuals will be resident in the community throughout the project's duration. These CCPs will serve as the liaison between the project team and the project community. They will assist in carrying out project sensitization, distributing of the project C2SSF, and posting C2SSP materials around the community after the team has departed. An inception workshop will be held to interact with and seek the opinions of key stakeholders regarding the project's purpose and procedures, which will be vital to secure community buy-in and for its successful execution.

*Intervention communities' mobilization and sensitization (Month 3&4)*. Following stakeholder engagement, mobilization and sensitization of communities, pilot in the intervention states will commence. This will be achieved by securing the support of the National NTDs Programme Office to collaborate with selected State NTDs Programme Officers. The project team will work with two state NTDs officers and two Local Government Area (LGA) NTDs officers, comprising the LGA's social mobilization officer and the LGA's NTDs focal person. The primary goal is to raise awareness about the project among community members, ensuring their active participation throughout its duration.

These initial engagement activities will be complemented by ongoing engagement throughout the project, promoting community ownership and allowing for adaptation to optimize project implementation. Additionally, there will be periodic advocacy visits to the project communities and continuous engagement with community stakeholders, both in-person and virtually, to ensure their involvement and commitment throughout the project, ultimately fostering community ownership.

During this community engagement and sensitization, the designed advertisements/notices will be produced after validation by the Federal Ministry of Health's national NTD programme office and deployed in intervention states as follows; **C2SSS**- to be deployed at strategic locations such as community centers; **C2SSP**- to be posted at primary, secondary and tertiary health facilities, commuter parks, religious centers, markets and schools; **C2SSF**- to be distributed at all public gatherings within the communities including the market squares, healthcare facilities, churches and schools. Additional adverts/notices will be placed on local Radio stations. These are aimed at adequately sensitizing the public on the procedures and their involvement in the process.

Control states will continue with the traditional surveillance systems without the intervention.

*Reception, collation, processing and analysis of photo and demographic data, and other data (Month 4–15).* Transmitted photos and demographic data will be received and collated in a central electronic data base and protected with an access code. Photos will be processed and sorted based on clarity, replications and complete data. Internet Protocol (IP) of the devices used in the transmission of the photos will be checked to eliminate repeated submissions. All data will be anonymized so that they will not be linked to individuals. A data protection expert will be involved to ensure compliance with relevant data protection laws. Initially, we will work with our expert consultants to identify, confirm, and match crowdsourced images to specific NTDs, while work on development, training and validation of the AI/ ML algorithm is ongoing. As soon as the AI/ ML algorithm has been trained and validated to the extent of confidence required to screen and match transmitted images, it will then be applied to replace identification/ matching of images by expert consultants. Subsequently, therefore, the AI/ML will automatically identify and match crowdsourced photos to specific NTDs. Arc Map, a digital mapping software will thereafter be used to map the respective target NTDs using the identified/ matched photos together with the location data along with the GPS location imprints of the transmitted images. Demographic and geographic data will be analysed using statistical software packages like SPSS and R to determine geographic and demographic distribution of these NTDs. Towards the end of the photo or image data collection, quantitative and qualitative techniques will be used to collect data from the subjects on their experience, acceptability, usability and feasibility of this crowdsourced image-based NTDs surveillance method. Data will also be collected from officers at local government NTDs programme offices on their experiences with the conventional NTDs surveillance method and possible perceptions about this new method.

*Feedback on preliminary diagnosis and information on actions to take (Personalized/ Individual Care) (Month 4–15).* Concurrently with data reception, transmission, collation and processing, there will be feedback to subjects whose data were transmitted through the telephone numbers used in transmitting such data. This will be done as the respective photos are successfully matched to specific NTDs. The feedback will be on the preliminary diagnosis of their conditions, counselling, and provision of information on appropriate actions to take as well as follow up. This will be undertaken by the project case managers and officers who are physicians with experiences regarding NTDs. They will also visit the communities periodically to provide diagnosis physically and support to identified subjects. In conjunction with the FMOH's national NTDs programme and local authorities/ health facilities as well as the community contact persons, standard operating procedure (SOPs) for investigating, triaging between the Federal, State, Local Government NTDs control/ elimination programme offices and responding to NTDs cases that are confirmed remotely will be developed to ensure confirmatory diagnosis and appropriate treatment is provided. This will involve leveraging on existing NTDs control/ elimination structures which includes frontline health facilities in every Local Government where identified cases of NTDs will be managed and counselled, and referral to tertiary health facilities where more serious cases beyond the capacity of the frontline facilities will be handled. Community contact persons together with selected community health workers at the local or community levels will additionally be trained as adherence counselors to track individuals who transmitted images and received feedback to ensure that they take recommended actions and assist in delivering specific messages against stigmatization.

*Large volume collection of photographs of manifestations of NTDs and development of artificial intelligence (AI) driven matcher of images to specific NTDs (Month 4–7).* This activity will be conducted simultaneously with the project implementation in intervention communities (month 3&4). In the initial stages of the project, expert consultants with extensive experience in NTDs research, along with officials from the National NTDs Programme Office under the

Federal Ministry of Health and Social Welfare as well as personnel from partner NGOs, will be requested to gather, validate, and categorize a substantial volume of photographs illustrating various stages of NTD manifestations prevalent in the project locations. Digital cameras will be employed for this purpose. Additionally, we will collaborate with researchers from universities, clinicians, and recognized NTDs experts in Nigeria and other parts of the world to solicit their contributions of confirmed NTDs cases' photographs from their diverse image collections. These collected photographs will form the dataset essential for creating and training the AI and Machine Learning (ML) algorithms. This algorithm when developed will facilitate the automated identification and matching of electronically transmitted crowdsourced images with specific NTDs. Furthermore, this dataset, combined with a comprehensive knowledge database will serve as the foundation for developing a point-of-care differential diagnosis and management mobile application.

Before applying the AI/ML photo matching algorithm to crowdsourced images, we will rigorously test it on a set of test images and compare its results with the outcomes of expert matching processes to ensure its validity and diagnostic accuracy.

## Study indicators

At the conclusion of the pilot study, our primary focus will be on evaluating the effectiveness and efficiency of the CIMS-NTDs method in comparison to the conventional surveillance approach. To assess this, we will primarily examine the quantity and quality of NTDs data collected using the CIMS-NTDs method from the pilot study site. This data will then be compared with the NTDs prevalence information gathered through the traditional surveillance method, both within the pilot study locations and in the control locations. It is important to note that this comparison will exclusively consider NTDs data collected from both methods during the study period.

The data collected for this study will primarily encompass the following:

1. Photographs: These will be visual representations of NTDs manifestations (morbidity images) submitted by participants, providing valuable visual information for diagnosis.

2. Metadata: Accompanying information, such as timestamps, location data, and patient details, will help contextualize and organize the photographs.

3. Qualitative data from NTDs stakeholders in study locations

4. Data assessing the perception of NTDs subjects and the people in the study locations about their perception about CIMS-NTDs and the ease of reporting using the CIMS-NTDs method.

## Data management and analysis plans

The majority of the data generated by the study will consist of photographs depicting various manifestations submitted by participants via WhatsApp or Telegram, along with accompanying metadata. Upon receipt, this data will be processed in our database, undergoing cleaning and organization before being transmitted to our experts for precise matching to specific NTDs. Stringent security measures will be in place to safeguard this data within the database. Upon the return of annotated results, they will be integrated into another section of the database, where they will be transformed into results and presented as preliminary diagnoses.

The data analysis plan, which encompasses the utilization of received photo and accompanying metadata to develop incidence and prevalence data for NTDs in our study sites. This data will be disaggregated by age, sex, educational attainment, and occupation of the subjects, providing a comprehensive overview of the affected demographics. Additionally, we will

gather information regarding the current management of NTDs among the subjects and present this information in a straightforward frequency distribution table.

To assess the efficacy of the CIMS-NTDs method compared to the traditional surveillance approach, we will collect NTDs prevalence data from control states and traditional NTDs surveillance prevalence data from intervention states. Our comparative analysis will consider various indicators, including but not limited to:

1. Proportion of suspected NTD cases (per disease) reported weekly/monthly.

2. Rate and timeliness of reporting of suspected cases.

3. Proportion of identified cases that received a response in the intervention states versus the control states.

4. Promptness of the response received.

5. Demographic and geographic distributions of identified cases per specific focus NTDs.

For geographic analysis, we will employ Arc GIS to create spatial maps highlighting NTDs morbidity hotspots within the pilot study states. These maps will be generated based on the NTDs images provided by the affected individuals and their respective communities.

In addition to quantitative data, we will gather qualitative insights through key informant interviews with NTDs stakeholders in the pilot states. These interviews will focus on their experiences with the traditional NTDs surveillance method and their perceptions of the CIMS-NTDs method. Furthermore, we will assess community members' experiences and perceptions of the CIMS-NTDs method using structured, interviewer-administered questionnaires. Responses will be compiled, aggregated, and visually presented in charts for clarity. Experiences will be categorized as either "good," "poor," or "neutral," while perceptions will be reported as either "positive" or "negative."

This comprehensive data analysis plan will enable us to evaluate the effectiveness of the CIMS-NTDs method and its potential benefits over traditional surveillance approaches, ultimately contributing to the enhancement of NTDs control and management strategies.

## Ethical considerations and informed consent

Ethical approval (NHREC Approval Number NHREC/01/01/2007-16/01/2023) was given by the National Health Research Ethics Committee (NHREC), Federal Ministry of Health, Abuja Nigeria. Also, approval was obtained from the Institutional Review Board of the Federal University of Technology, Owerri, Nigeria. Our advertisements will bear a brief description of the project, its intentions and the role of potential participants in lay language, and request that only those who give consent to participate should transmit images of NTDs manifestations (without their faces) and accompanying demographics (without their names). To further validate informed consent, there will be an automated response from our WhatsApp and Telegram platforms prompting those who transmit images/ data to confirm their consent and permission to use their data in this research. Additionally, our community contact persons who will assist those without smart devices to take photographs of the images and transmit will collect informed written consent from such subjects. For the children and vulnerable groups, their parents and care givers will be allowed to provide the consent on their behalf with the children assenting to it. Hard copies of signed consent forms will be collected from our Community Contact Persons and stored in a secure steel cabinet while electronic consent will be stored in our password secured electronic data base. Data collection and analysis method will be in conformity with the terms and conditions for source data.

## Discussion

This protocol presents a comprehensive description of an ongoing pilot study introducing an innovative surveillance method for Neglected Tropical Diseases (NTDs), aligning with the WHO's NTDs elimination agenda for 2030, as outlined in the NTDs Roadmap 2021–2030 [5]. The surveillance method employed in this study breaks new ground by leveraging crowdsourcing and smartphone technology to empower community members. Participants are sensitized to capture clear images of NTDs manifestations and transmit them, along with their complete demographic data including their age, location, sex, duration of manifestation and contact phone number for the purpose of feedback provision, via platforms such as WhatsApp or Telegram. This approach not only ensures the privacy of individuals reporting NTDs but also mitigates the potential stigmatization and shame often associated with visiting healthcare facilities for NTDs reporting [13]. Furthermore, it actively engages subjects in the surveillance process, promoting inclusivity within the population. Simultaneously, it facilitates the surveillance of multiple NTDs, which is especially valuable for post-elimination monitoring [14].

A notable limitation of this project is the restricted access to smartphones and other smart devices among many residents in rural communities, which are often the areas most heavily affected by NTDs [15]. To address this limitation, a community contact person (CCP) is selected, trained, and provided with a smartphone if the individual does not have one. The CCP plays a pivotal role in taking photographs and transmitting them on behalf of individuals who wish to report NTDs cases but lack the necessary technology.

Dissemination of the knowledge and information acquired from the project will occur through diverse mechanisms. First, a dissemination seminar will be conducted to communicate project outcomes and information to stakeholders. Second, policy briefs will be developed and shared with decision-makers to facilitate timely implementation and decision-making [16]. Third, the findings will be made publicly available through scientific publications in open-access peer-reviewed journals and presentations at local and international scientific conferences. Fourth, project outcomes and information will be accessible on our organizational website, the Bill and Melinda Gates Foundation website, as well as the websites of other partner organizations. Additionally, key outcomes, outputs, and takeaways from the project will be shared on social media platforms such as Facebook, Twitter, and Instagram to enhance public awareness and engagement.

It is anticipated that this project phase will yield a timely, cost-effective, population-inclusive, feasible, and efficient alternative to traditional NTDs surveillance methods. It holds promise for scalability, integration into ongoing NTDs elimination efforts and health systems, enhanced large-scale surveillance of NTDs in the country and the sub-Saharan African region, simultaneous monitoring of multiple NTDs, improved diagnosis, case management, prevention, and, ultimately, significant contributions toward NTDs elimination. Furthermore, this method will prove invaluable for post-elimination surveillance, enabling the prompt identification of residual cases or underserved areas and groups to prevent resurgence. Future endeavors will explore full automation and digitalization of the entire image-based surveillance and response process with minimal human intervention, utilizing artificial intelligence, machine learning, cloud computing tools, and further personalization for affected subjects.

In conclusion, this pilot study's innovative approach to NTDs surveillance holds promise to significantly improve the surveillance for NTDs and accelerating the WHO's attainable goals for NTDs elimination by 2030. Despite challenges related to smartphone accessibility in rural areas, the inclusion of CCPs helps bridge this gap, ensuring equitable participation in the surveillance process.

## Supporting information

**S1 File.**

(ZIP)

## Acknowledgments

We thank the Nigeria Federal Ministry of Health and Social Welfare NTDs Programme Office for their collaboration and support.

## Author Contributions

**Conceptualization:** Uchechukwu Madukaku Chukwuocha.

**Data curation:** Christopher Sule Oyamienlen, Ayoola Oluwaseun Bosede.

**Formal analysis:** Christopher Sule Oyamienlen, Ayoola Oluwaseun Bosede.

**Funding acquisition:** Uchechukwu Madukaku Chukwuocha.

**Investigation:** Christopher Sule Oyamienlen, Ayoola Oluwaseun Bosede.

**Methodology:** Uchechukwu Madukaku Chukwuocha.

**Project administration:** Uchechukwu Madukaku Chukwuocha, Christopher Sule Oyamienlen.

**Resources:** Uchechukwu Madukaku Chukwuocha.

**Supervision:** Uchechukwu Madukaku Chukwuocha.

**Validation:** Ikechukwu Nosike Dozie.

**Visualization:** Christopher Sule Oyamienlen.

**Writing – original draft:** Uchechukwu Madukaku Chukwuocha, Christopher Sule Oyamienlen, Ayoola Oluwaseun Bosede.

**Writing – review & editing:** Ikechukwu Nosike Dozie.

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
