## [Decision Letter · Decision Letter 0]

10 Nov 2023

PONE-D-23-31987Crowdsourced Image-Based Morbidity Hotspot Surveillance for Neglected Tropical Diseases (CIMS-NTDs): Study ProtocolPLOS ONE

Dear Dr. Chukwuocha,

Thank you for submitting your manuscript to PLOS ONE. After careful consideration, we feel that it has merit but does not fully meet PLOS ONE’s publication criteria as it currently stands. Therefore, we invite you to submit a revised version of the manuscript that addresses the points raised during the review process.

We look forward to receiving your revised manuscript.

Kind regards,

Clement Ameh Yaro, Ph.D

Academic Editor

PLOS ONE

Journal Requirements:

2. In your Methods section, please include additional information about your dataset and ensure that you have included a statement specifying whether the collection and analysis method complied with the terms and conditions for the source of the data.

3. Thank you for stating the following in the Acknowledgments Section of your manuscript: "We would like to express our heartfelt gratitude to the following organizations and entities for their invaluable support and contributions to this research project;

The Bill and Melinda Gates Foundation for their generous funding, which made this study possible. The Nigeria Federal Ministry of Health NTD Programme Office for their collaboration and partnership throughout the project. "

Please remove any funding-related text from the manuscript and let us know how you would like to update your Funding Statement. Currently, your Funding Statement reads as follows: "UMC received the grant. The grant number is INV-048721. The funder is Bill and Melinda Gates Foundation https://www.gatesfoundation.org/

The funder did not play any role in the design, analysis, decision to publish or preparation of the manuscript."

Reviewers' comments:

Reviewer's Responses to Questions

**Comments to the Author**

1. Does the manuscript provide a valid rationale for the proposed study, with clearly identified and justified research questions?

Reviewer #1: Yes

Reviewer #2: Yes

2. Is the protocol technically sound and planned in a manner that will lead to a meaningful outcome and allow testing the stated hypotheses?

Reviewer #1: Yes

Reviewer #2: Yes

3. Is the methodology feasible and described in sufficient detail to allow the work to be replicable?

Reviewer #1: Yes

Reviewer #2: Yes

4. Have the authors described where all data underlying the findings will be made available when the study is complete?

Reviewer #1: No

Reviewer #2: Yes

5. Is the manuscript presented in an intelligible fashion and written in standard English?

Reviewer #1: Yes

Reviewer #2: Yes

6. Review Comments to the Author

You may also provide optional suggestions and comments to authors that they might find helpful in planning their study.

Reviewer #1: Check acronym: CCP vs CPP

This study if done well promises to be of great value in rural and hard to reach populations where NTDs are endemic. Here are a few questions for clarity and some suggestions that could possibly improve on this study:

1.Are the CCPs the same as Community Health Workers? I notice you mention people who distribute ivermectin, what other role do these people play in the community? Will not this study distract them from their regular work.

2.In your discussion with the Health Ministry (at all levels) where these community personnel going to be volunteering their service or they will get additional duties with this project? This is important to think about as uncooperating community workers might hamper your study.

3.NTDs affect very poor communities. To have a cell phone with no data will not help this study. Have you engaged Internet Service Providers (ISP) to be part of the study? This was possible during the Covid-19 era, with ISPs getting on board particularly with rural students to give them access to academic institutions teaching and learning material. Perhaps when the person contacts the number you gave them for notifying your office, the login triggers the internet data for them, and be locked again as soon as they are off your repository. By this you will be targeting the SDG17 on partnership, forming a Private-Public Partnership arrangement.

4.If the suggestion in point 3 above is not possible, let them use free WiFi spots in the area, also this can be provided by the government of ISP.

5.For your standard, Photo Literature search (line 211), I would think the WHO is the most relevant place to consider as your first port of call before going to other sources. This could cut your time for phase 2

6.Match the sites (intervention and control) as close as possible, in order to address what you wrote in lines 144-145, “these methods prove ineffective in circumstances where access to NTD-endemic areas is restricted, such as during the COVID-19 pandemic, civil unrest, or natural disasters”.

This way you could eliminate any bias that could influence the outcomes in your study

7.In the intervention sites, both methods are going to continue concurrently, how are you going to manage the influence from your study on the traditional method of surveillance. For instance, an affected person can opt to report the case to both your study and to the Health Ministry at the same time. Will you take this influence into consideration when you analyse the data?

8.When the study is completed and you are publishing the results, are you going to allow other researchers and clinicians access to your database as is a requirement for PLOS data policy?

Reviewer #2: This is a very nice manuscript, that describes the study protocol of an ongoing and very important piece. I have made significant comments within the text most around grammatical flaws, restructuring of some sections, rephrasal of several sentences amongst others. Additionaly, I think too much of details have been provided in the supplementary section. Authors should only provide the ethical approval letter, and remove the others

7. PLOS authors have the option to publish the peer review history of their article (what does this mean?). If published, this will include your full peer review and any attached files.

Reviewer #1: **Yes: **Dr Kuku Voyi

Reviewer #2: **Yes: **Mogaji Hammed

---

## [Author Response · Author response to Decision Letter 0]

7 Dec 2023

RESPONSE TO ACADEMIC EDITOR AND THE REVIEWRS

Section Point Raised Response

Abstract; Introduction This introduction is quite lengthy. Authors should summarize in max 5 lines.

I would expect that the authors would clearly express that this study aims to develop and test a new methodology The abstract introduction has been summarized and re-presented in a more succinct manner

Abstract; Methods Authors also need to re-write/re-structure this section; with clear details about the study area, participants, study design/approach, timings, brief details of the newly developed tool, and how it would be used, details on the evaluation protocol; what the expected results would be and the likes. The methods section of the abstract has been re-presented using the prompts as suggested by the reviewer. 

Abstract; Ethics and Dissemination Authors should re-write this as well, 1/2 lines for ethics

and a separate sub-section for funding.

e.g., Funding: This study received financial support from XYZ (Grant number: XYZ) This sub-section has been re-introduced as Ethical Approval with a stand-alone sub-section for funding.

Introduction; line 66 Not all of them. So kindly rephrase this Some of the NTDs are marked for elimination, some for control, some for eradication while some for elimination as a public health problem. The statement has be re-written to reflect this.

Introduction; line 71 could these come up before line 66-71 The paragraph truly sounds like it should introduce the subject matter. It has been brought up to begin the introduction. 

Introduction; line 85 It is important to emphasize the challenges that necessitate the prose on traditional surveillance. The challenges are the problem with the traditional NTDs surveillance method. This has been elaborated and brought up before line 85.

Introduction; line 87 this should as well be referenced A proper citation has been attached to the statement

Introduction; line 89 this should come up around line 85, and should be referenced A proper citation has been provided and the statement has been moved to line 85.

Introduction; line 102 this can start as a new paragraph, and of course rephrased:

consider staring like; The authors agree with this argument. We have moved it to a new paragraph and we have also rephrased the statement.

Introduction; line 105 This could have been the first sub-section under the methodology section.

"Development of XYZ" This statement is introducing our study objectives as part of our introduction. We believe that that is where it should be. For clarity’s sake, we have rephrased the statement.

Introduction; line 106 This could have been the first sub-section under the methodology section.

"Development of XYZ" We also believe that this is still part of our introduction which was later elaborated in the methodology. We have also rephrased the statement for better understanding.

Introduction; line 111 this entire section needs to be rephrased and reported in a more succinct style. We have rephrased and presented in a more succinct style.

Introduction; line 115 If the images were uploaded by third-parties on behalf of affected persons, since most affected persons may live in resource constrained settings, where access to smart-phones, electricity and internet may be lacking. Then one would wonder who gets the feedback proposed here 

Then the line 119-121, where the authors mentioned that hygiene practices can also be provided is confusing; authors should be very specific on what would or would not be implemented Though in our methods section, this objective is elaborated, for clarity, we have added additional statement that shows that the CCPs who obtains and forward NTDs manifestation photographs on behalf of the subject will also be employed to take feedback to the subject. The CCPs will be trained in basic counselling provision for home management of common NTDs.

Regarding hygienic practices. The subject will be counselled and trained on basic home care of NTDs like elephantiasis and trachoma that requires regular cleaning with warm water and antiseptic disinfectants.

Introduction; line 122 This section is also not clear enough. What exactly would The section has been re-written. The objective is the development and AI/ML algorithm model that will help in the prompt and accurate identification and annotation of morbidity images that will be fed into it to specific NTDs, providing a sort of differential diagnosis. Thus, eliminating the need for human annotators as we currently have.

Introduction; line 128 This section should also go into the last paragraph of the discussion Even though the protocol manuscript has no discussion section. The statement truly seems like a good concluding statement, we have therefore, moved it to the very last paragraph of the manuscript, after the statement of how the research finding will be disseminated.

Materials and Methods; line 142 this entire section should serve as the justification for developing the novel methods and should fit into the introductory section (the last paragraphs that leads to the aims/objective of the study) We are not too sure that the PLOS ONE protocol manuscript for publication should contain a justification section; in the interim, we have created a justification sub-section under the introduction and we have moved the statement therein because we agree that the statement is actually justifying the need for the study.

Materials and Methods; line 176 Needs to be referenced Appropriate reference has been provided.

Materials and Methods; line 187 Please rephrase this The statement has been re-presented in a clearer form.

Materials and Methods; line 190 this is the first mention of this acronym

 NGOs like WHO, UNICEF etc are common acronyms that often time do not require interpretation in reports. The acronym has however, been written in full.

Materials and Methods; line 192 The study design is still not clear. Author should used a flow chat to describe the implementation process; in the intervention and cotrol states, who would be the target audience, and how will the interventions be deployed. 

 Kindly refer to the “Description of Processes” section in line 198 which has now been re-named as “Study Design”. Also refer to the project’s comprehensive flowchart (Fig 1) in Appendix

Materials and Methods; line 198 This should be merged with the study design section which I suggested above A study design section has been created and this has been made a sub-section under the newly created study design section

Materials and Methods; line 306 Throughout this process, it is still not clear how the NTDs photographs would be sourced/obtained from the community

 The process of photograph collection was described in line 208 – 245 where we described our advertisement materials. We then give the description of our study sites in line 247 – 270. We explained that advert notice will be put out in our study sites calling for NTDs subjects and their community members to send us photos of NTDs manifestations upon notice through our dedicated WhatsApp and Telegram channels. We also described the process of community entry, engagement and sensitization and the selection of CCPs within the people that will help us to get manifestation photographs of subjects who cannot send us photographs on their own.

Materials and Methods; line 308 this section should be described separately; the first section on how to manage the photographs received; and the second section on the AI development etc

 This activity will commence in the 4th month of our project inception. It will continue simultaneously with other activities including morbidity images crowdsourcing from the communities. We are presenting it here because the entire process was described within their timelines of implementation.

Materials and Methods; line 327 this should be discussed simultaneously with other section, and not separately as described here. The section has been cut out and joined positioned directly under the section that describes the Intervention communities' mobilization and sensitization

Data Management Plane; line 384 This is the first time of mentioning this, I think the detailed methodology and platform should have been described earlier before getting to this stage

 This is not the first time we mentioned WhatsApp and Telegram in the protocol, it was first mentioned in the Abstract, then under study aim that has been changed to Justification then it was mentioned under “description of process”

Data Management Plane; line 391 Authors could merge this section with the Outcome Measure section, and title the section as "Study Indicators". Under this section they should describe the data, and also the explanatory and outcome variable, and also the planned statistical analysis 

 They have been merged.

Ethical considerations and informed consent; line 432 Please delete this, there is no reason to have the protocol number in this section

 Protocol number has been removed from the statement.

Discussion; line 455 this should be specified; what demographic data; sex, age, name or what The types of demographic data that will be obtained has been clearly spelt out.

Discussion; line 457 on the long run, would this be achieved? when providing care to this population, the healthcare worker would still be aware. Are there reports of stigmatization from HCW to patients or among community members and those affected Note that the care that each subject will receive based on their specific diagnosis will be personalized and likely to be delivered to in the homes of these subjects. The stigmatization here in most cases does not emanate from the HCW but from other members of the community. The CIMS-NTDs method prevent the situation where a subject is made to wait endlessly at the health center trying to access care.

Discussion; line 466 what measures of This comment is not clear. The acronym there is CCP (Community Contact Person) and not CPP.

Discussion; line 478 to Correction taken

ACADEMIC REVIEWERS COMMENT

Comment 1 Please ensure that your manuscript meets PLOS ONE's style requirements, including those for file naming The manuscript has been re-designed to meet the PLOS ONE’s style

Comment 2 In your Methods section, please include additional information about your dataset and ensure that you have included a statement specifying whether the collection and analysis method complied with the terms and conditions for the source of the data. The Methods section has been enhanced to include additional information about our dataset. Our data collection and analysis method will be in conformity with the terms and conditions for source data. We have included a statement indicating this under our ethical consideration and informed consent section 

Comment 3 Thank you for stating the following in the Acknowledgments Section of your manuscript: "We would like to express our heartfelt gratitude to the following organizations and entities for their invaluable support and contributions to this research project;

The Bill and Melinda Gates Foundation for their generous funding, which made this study possible. The Nigeria Federal Ministry of Health NTD Programme Office for their collaboration and partnership throughout the project.

Please remove any funding-related text from the manuscript The funding-related texts has been removed from the manuscript

Comment 4 In your Data Availability statement, you have not specified where the minimal data set underlying the results described in your manuscript can be found. PLOS defines a study's minimal data set as the underlying data used to reach the conclusions drawn in the manuscript and any additional data required to replicate the reported study findings in their entirety. All PLOS journals require that the minimal data set be made fully available. This is a proposal of an ongoing project; we are yet to generate or analyse any data that may yield a valid conclusion. No result was described in the proposal manuscript.

---

## [Decision Letter · Decision Letter 1]

27 Mar 2024

PONE-D-23-31987R1Protocol for the Crowdsourced Image-Based Morbidity Hotspot Surveillance Method for Neglected Tropical Diseases (CIMS-NTDs)PLOS ONE

Dear Dr. Chukwuocha,

Thank you for submitting your manuscript to PLOS ONE. After careful consideration, we feel that it has merit but does not fully meet PLOS ONE’s publication criteria as it currently stands. Therefore, we invite you to submit a revised version of the manuscript that addresses the points raised during the review process.

We look forward to receiving your revised manuscript.

Kind regards,

Clement Ameh Yaro, Ph.D

Academic Editor

PLOS ONE

Journal Requirements:

Reviewers' comments:

Reviewer's Responses to Questions

**Comments to the Author**

1. Does the manuscript provide a valid rationale for the proposed study, with clearly identified and justified research questions?

Reviewer #1: Yes

Reviewer #2: Yes

2. Is the protocol technically sound and planned in a manner that will lead to a meaningful outcome and allow testing the stated hypotheses?

Reviewer #1: Yes

Reviewer #2: Yes

3. Is the methodology feasible and described in sufficient detail to allow the work to be replicable?

Reviewer #1: Yes

Reviewer #2: Yes

4. Have the authors described where all data underlying the findings will be made available when the study is complete?

Reviewer #1: No

Reviewer #2: No

5. Is the manuscript presented in an intelligible fashion and written in standard English?

Reviewer #1: No

Reviewer #2: Yes

6. Review Comments to the Author

You may also provide optional suggestions and comments to authors that they might find helpful in planning their study.

**Reviewer #1**: This revised version requires language editing

As the previous review comments were suggestions to improve the study, the authors have selected what feels right for them.

The writing of this protocol flows better in the arranged format.

Check these and correct:

References are not sequential in the Introduction section, in Line 62 you started with 1,3,4 and only in line 78 do you cite reference 2.

Social media handles mentioned in line 107, the rest of the protocol is mentioning WhatsApp and Telegram. Does this mean other social media will be considered in future? Mention that in line 107.

The Pilot sites are 6/36 of states in Nigeria, include this information in the demographics paragraph to give the reader a clear perspective of the size of the country. Are all 36 states NTDs endemic?

It is not clear how this data is going to be managed, there is something about IP, but I think some explanation is required to clarify this point

**Reviewer #2:** I believe the authors have satisfactorily responded to all the comments raised earlier on.

However, they have introduced a new section on Justification. my initial suggestion to them was to just include a few lines to justify the protocol, as part of their introduction. I guess this must have been wrongly understood. Authors should try to reconcile, and the lines in the new justification section are kindof repetitive of those earlier raised in the introduction.

7. PLOS authors have the option to publish the peer review history of their article (what does this mean?). If published, this will include your full peer review and any attached files.

Reviewer #1: **Yes: **Kuku Voyi

Reviewer #2: **Yes: **HAMMED OLADEJI MOGAJI

---

## [Author Response · Author response to Decision Letter 1]

2 Apr 2024

RESPONSES TO REVIEWERS' COMMENTS

Reviewer 1

General 

Comment

This revised version requires language editing 

Response

The manuscript has been subjected to English language editing, it was improved for grammar, spellings, punctuations and clarity.

Comment

As the previous review comments were suggestions to improve the study, the authors have selected what feels right for them.

Response

Our review was made in the light of the recommendations from the academic editors and the two peer reviewers. We strictly followed the reviewers comments to carry out the revision of the manuscript. We have made effort to make the manuscript as smooth flowing as possible through further English language editing which was done by an English expert.

References 

Comment

References are not sequential in the Introduction section, in Line 62 you started with 1,3,4 and only in line 78 do you cite reference 2. 

Response

It was an oversight from the editing author, the error has been corrected. The references are now properly presented in the order of appearances.

Introduction 

Comment

Social media handles mentioned in line 107, the rest of the protocol is mentioning WhatsApp and Telegram. Does this mean other social media will be considered in future? Mention that in line 107. 

Response

We made use of the phrase social media handles to give a blanket name to WhatsApp and Telegram platform. These are the most common platforms that the Nigerian population are conversant with. We do not plan to include other social media handles in the future, we have therefore replaced the phrase “social media handles” with “WhatsApp and Telegram” 

Materials and methods

Comment

Study setting: The Pilot sites are 6/36 of states in Nigeria, include this information in the demographics paragraph to give the reader a clear perspective of the size of the country. Are all 36 states NTDs endemic? 

Response

All the 36 states in Nigeria plus the FCT have at least one NTD being endemic. 

A statement mentioning the number of states in Nigeria and the number of the pilot study sites have been included in the “Study setting” sub-section.

Comment

Description of process: It is not clear how this data is going to be managed, there is something about IP, but I think some explanation is required to clarify this point 

Response

Activities in the Reception, collation, processing and analysis of photo and demographic data, and other data (Month 4-15) starting from line 308 – 330 described the manners with which the received data will be managed. 

The use of IP number- internet protocol number of the device that will send the data to us is just to add another layer of security to the data management activities. The IP number of the devices that sends in photo-data will be identified to allow the researchers to know where the data is coming from, this will help us checkmate the situation of multiple submission of the same data thereby maintain the integrity of the submitted data. 

In addition to this, the section on “Data Management and Analysis Plans” also gave a succinct description of how data that will be received will be dealt with prior to forwarding to our experts for annotation and what will be done with the annotated image-data.

 Reviewer 2

Comment

I believe the authors have satisfactorily responded to all the comments raised earlier on.

However, they have introduced a new section on Justification. my initial suggestion to them was to just include a few lines to justify the protocol, as part of their introduction. I guess this must have been wrongly understood. Authors should try to reconcile, and the lines in the new justification section are kind of repetitive of those earlier raised in the introduction. 

Response

The JUSTIFICATION sub-heading has been collapsed into the INTRODUCTION section with the two merged together while avoiding duplication of points raised.

---

## [Decision Letter · Decision Letter 2]

22 Apr 2024

Protocol for the Crowdsourced Image-Based Morbidity Hotspot Surveillance Method for Neglected Tropical Diseases (CIMS-NTDs)

PONE-D-23-31987R2

Dear Dr. Chukwuocha,

We’re pleased to inform you that your manuscript has been judged scientifically suitable for publication and will be formally accepted for publication once it meets all outstanding technical requirements.

Kind regards,

Clement Ameh Yaro, Ph.D

Academic Editor

PLOS ONE

Additional Editor Comments (optional):

Reviewers' comments:

Reviewer's Responses to Questions

**Comments to the Author**

1. Does the manuscript provide a valid rationale for the proposed study, with clearly identified and justified research questions?

Reviewer #1: Yes

Reviewer #2: Yes

2. Is the protocol technically sound and planned in a manner that will lead to a meaningful outcome and allow testing the stated hypotheses?

Reviewer #1: Yes

Reviewer #2: Yes

3. Is the methodology feasible and described in sufficient detail to allow the work to be replicable?

Reviewer #1: Yes

Reviewer #2: Yes

4. Have the authors described where all data underlying the findings will be made available when the study is complete?

Reviewer #1: Yes

Reviewer #2: Yes

5. Is the manuscript presented in an intelligible fashion and written in standard English?

Reviewer #1: Yes

Reviewer #2: Yes

6. Review Comments to the Author

You may also provide optional suggestions and comments to authors that they might find helpful in planning their study.

Reviewer #1: In this revised manuscript,

1. The language has been edited and it reads well.

2. All requested changes have been done.

I have no further comments to the authors, well done

Reviewer #2: All comments have been sufficiently addressed

All comments have been sufficiently addressed"=

All comments have been sufficiently addressed

7. PLOS authors have the option to publish the peer review history of their article (what does this mean?). If published, this will include your full peer review and any attached files.

Reviewer #1: **Yes: **Kuku Voyi

Reviewer #2: **Yes: **MOGAJI HAMMED OLADEJI

---

## [Editor Report · Acceptance letter]

29 Apr 2024

PONE-D-23-31987R2 

PLOS ONE

Dear Dr. Chukwuocha, 

I'm pleased to inform you that your manuscript has been deemed suitable for publication in PLOS ONE. Congratulations! Your manuscript is now being handed over to our production team.

Kind regards, 

on behalf of

Dr. Clement Ameh Yaro 

Academic Editor

PLOS ONE